# Continual Audio-Visual Sound Separation

**Weiguo Pian**[1]  **Yiyang Nan**[2]  **Shijian Deng**[1]  **Shentong Mo**[3]  **Yunhui Guo**[1]  **Yapeng Tian**[1]

[1] The University of Texas at Dallas  [2] Brown University  [3] Carnegie Mellon University

## Abstract

In this paper, we introduce a novel continual audio-visual sound separation task, aiming to continuously separate sound sources for new classes while preserving performance on previously learned classes, with the aid of visual guidance. This problem is crucial for practical visually guided auditory perception as it can significantly enhance the adaptability and robustness of audio-visual sound separation models, making them more applicable for real-world scenarios where encountering new sound sources is commonplace. The task is inherently challenging as our models must not only effectively utilize information from both modalities in current tasks but also preserve their cross-modal association in old tasks to mitigate catastrophic forgetting during audio-visual continual learning. To address these challenges, we propose a novel approach named ContAV-Sep (**Cont**inual **A**udio-**V**isual Sound **Sep**aration). ContAV-Sep presents a novel Cross-modal Similarity Distillation Constraint (CrossSDC) to uphold the cross-modal semantic similarity through incremental tasks and retain previously acquired knowledge of semantic similarity in old models, mitigating the risk of catastrophic forgetting. The CrossSDC can seamlessly integrate into the training process of different audio-visual sound separation frameworks. Experiments demonstrate that ContAV-Sep can effectively mitigate catastrophic forgetting and achieve significantly better performance compared to other continual learning baselines for audio-visual sound separation. Code is available at: https://github.com/weiguoPian/ContAV-Sep_NeurIPS2024.

## 1 Introduction

Humans can effortlessly separate and identify individual sound sources in daily experience [25, 7, 64, 33]. This skill plays a crucial role in our ability to understand and interact with the complex auditory environments that surround us [34]. However, replicating this capability in machines remains a significant challenge due to the inherent complexity of real-world auditory scenes [7, 77]. Inspired by the multisensory perception of humans [62, 60], audio-visual sound separation tackles this challenge by utilizing visual information to guide the separation of individual sound sources in an audio mixture.

Recent advances in deep learning have led to significant progress in audio-visual sound separation [84, 23, 21, 67, 14, 65, 81, 63, 11, 71]. Benefiting from more advanced architectures (*e.g.,* U-Net [84, 23], Transformer [14], and diffusion models [27]) and discriminative visual cues (*e.g.,* grounded visual objects [67], motion [83], and dynamic gestures [21]), audio-visual separation models are able to separate sounds ranging from domain-specific speech, musical instrument sounds to open-domain general sounds within training sound categories. However, a limitation of these studies is their focus on scenarios where all sound source classes are presently known, overlooking the potential inclusion of unknown sound source classes during inference in real-world applications. This oversight leads to the *catastrophic forgetting* issue [32, 3], where the fine-tuning of models on new classes detrimentally impacts their performance on previously learned classes. Despite Chen et al. [14] demonstrating that their iQuery model can generalize to new classes well through simple fine-tuning, it still suffers from the catastrophic forgetting problem on old classes. This prevents the trained models from

continuously updating in real-world scenarios, impeding their adaptability to dynamic environments. The question *how to effectively leverage visual guidance to continuously separate sounds from new categories while preserving separation ability for old sound categories* remains open.

To bridge this gap, we introduce a novel *continual audio-visual sound separation* task by integrating audio-visual sound separation with continual learning principles. The goal of this task is to develop an audio-visual model that can continuously separate sound sources in new classes while maintaining performance on previously learned classes. The key challenge we need to address is catastrophic forgetting during continual audio-visual learning, which occurs when the model is updated solely with data from new classes or tasks, resulting in a significant performance drop on old ones. We illustrate our new task and the catastrophic forgetting issue in Fig. 1.

Unlike typical continual learning problems such as task-, domain-, or class-incremental classification in visual domains [2, 57, 38, 53, 85], which result in progressively increasing logits (or probability distribution) across all observed classes at each incremental step, our task uniquely produces fixed-size separation masks throughout all incremental steps. In this context, each entry in the mask does not directly correspond to any specific classes. Additionally, the new task involves both audio and visual modalities. Therefore, simply applying existing visual-only methods cannot fully exploit and preserve the inherent cross-modal semantic correlations. Very recently, Pian *et al.* [53] and Mo *et al.* [44] extended continual learning to the audio-visual domain, but both focused on classification tasks.

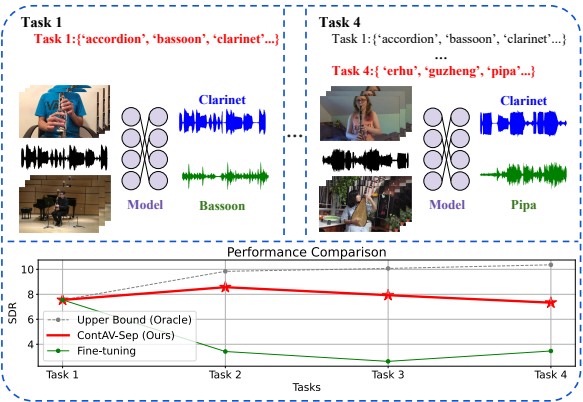

Figure 1: **Top**: Illustration of the continual audio-visual sound separation task, where the model (separator) learns from sequential audio-visual sound separation tasks. **Bottom**: Illustration of the catastrophic forgetting problem in continual audio-visual sound separation and its mitigation by our proposed method. Fine-tuning: Directly fine-tune the separation model on new sound source classes; Upper bound: Train the model using all training data from seen sound source classes.

To address these challenges, in this paper, we propose a novel approach named ContAV-Sep (**Cont**inual **A**udio-**V**isual Sound **Sep**aration). Upon the framework, we introduce a novel *Cross-modal Similarity Distillation Constraint (CrossSDC)* to not only maintain the cross-modal semantic similarity through incremental tasks but also preserve previously learned knowledge of semantic similarity in old models to counter catastrophic forgetting. The CrossSDC is a generic constraint that can be seamlessly integrated into the training process of different audio-visual sound separators. To evaluate the effectiveness of our proposed ContAV-Sep, we conducted experiments on the MUSIC-21 dataset within the framework of continual learning, using the state-of-the-art audio-visual sound separation model iQuery [14] and a representative audio-visual sound separation model Co-Separation [23], as our separation base models. Experiments demonstrate that ContAV-Sep can effectively mitigate catastrophic forgetting and achieve significantly better performance than other continual learning baselines. In summary, this paper contributes follows:

**(i)** To explore more practical audio-visual sound separation, in which the separation model should be generalized to new sound source classes continually, we pose a *Continual Audio-Visual Sound Separation* task that trains the separation model under the setting of continual learning. To the best of our knowledge, this is the first work on continual learning for audio-visual sound separation.

**(ii)** We propose ContAV-Sep for the new task. It uses a novel cross-modal similarity distillation constraint to preserve cross-modal semantic similarity knowledge from previously learned models.

**(iii)** Experiments on the MUSIC-21 dataset can validate the effectiveness of our ContAV-Sep, demonstrating promising performance gain over baselines.

## 2 Related Work

**Audio-Visual Sound Separation.** Audio-visual sound separation aims to separate individual sound sources from an audio mixture guided by visual cues. A line of research emerges under various scenarios, such as separating musical instruments [23, 84, 79, 21, 83, 67], human speech [20, 1, 18, 49, 15], or sound sources in in-the-wild videos [22, 70]. Many frameworks and methods have been proposed to address challenges within specific problem settings. For instance, the extraction of face embeddings proves beneficial for speech audio separation [18]. Moreover, incorporating object detection can provide an additional advantage [23, 22]. The utilization of trajectory optical flows to leverage temporal motion information in videos, as demonstrated by [83], also yields improvements. In this work, not competing on designing stronger separators, we would advance the exploration of the audio-visual sound separation within the paradigm of continual learning. We investigate how a model can learn to consistently separate sound sources from sequential separation tasks without forgetting previously acquired knowledge.

**Continual Learning.** The field of continual learning has drawn significant attention, especially in visual domains, with various approaches addressing this challenge. Notable among these are regularization-based methods, exemplified in works such as [32, 3, 31, 39]. These approaches involve applying regularization to crucial parameters associated with old tasks to maintain the model's capabilities and during incremental steps, less important parameters are given higher priority for updates compared to important ones. Conversely, several works [57, 9, 5, 26, 12, 54, 8, 10, 40] applied rehearsal-based pipelines to enable the model review previously learned knowledge. For instance, Rebuffi *et al.* [57] proposed one of the most representative exemplars selection strategy Nearest-Mean-of-Exemplars (NME), selects the most representative exemplars in each class based on the distance to the feature center of the class. Meanwhile, pseudo-rehearsal [47, 48, 66] employs generative models to create pseudo-exemplars based on the estimated distribution of data from previous classes. Moreover, architecture-based/dynamic architecture methods[52, 4, 46, 24, 28, 37, 40] proposed to modify the model architecture itself to enable the model to acquire new knowledge while mitigating the forgetting of old knowledge. Specifically, Pham *et al.* [52] proposed a dual network architecture, in which one is to learn new tasks while the other one is for retaining knowledge learned from old tasks. Wang *et al.* [72] combined the dynamic architecture and distillation constraint to mitigate the issue of continual-increasing overhead problem in dynamic architecture-based continual learning method. However, above studies mainly concentrate on the area of continual image classification. Recently, researchers also explored other continual learning scenarios beyond image classification. For instance, Park *et al.* [50] extend the knowledge distillation-based [2, 17] continual image classification method to the domain of video by proposing the time-channel distillation constraint. Douillard *et al.* [16] proposed to tackle the continual semantic segmentation task with multi-view feature distillation and pseudo-labeling. Xiao *et al.* [78] further addressed the continual semantic segmentation problem through weights fusion strategy between old and current models. Wang *et al.* [75] addressed the continual sound classification task through generative replay. Furthermore, continual learning has also been explored in the domain of language/vision-language learning tasks [30, 43, 59, 61, 19, 86], self-supervised representation learning [19, 42, 55, 80, 35, 74], audio classification [6, 73] and fake audio detection [41, 82], etc. Despite the success of existing continual learning methods in various scenarios, their applicability in the domain of continual audio-visual sound separation is still unexplored. Although Pian *et al.* [53] and Mo *et al.* [44] proposed to tackle the catastrophic problem in audio-visual learning, their studies mainly concentrated in the area of audio-visual video classification. In contrast to existing works in continual learning, in this paper, we delves into the continual audio-visual sound separation, aiming to tackle the challenge of catastrophic forgetting specifically in the context of separation mask prediction for complicated mixed audio signals within joint audio-visual modeling.

## 3 Method

### 3.1 Problem Formulation

**Audio-Visual Sound Separation.** Audio-visual sound separation aims to separate distinctive sound signals according to the given associated visual guidance. Following previous works [14, 23, 67, 21, 79], we adopt the common "mix-and-separation" training strategy to train the model. Given two videos $V_1(s_1, v_1)$ and $V_2(s_2, v_2)$, we can obtain the input mixed sound signal $S$ by mixing two

video sound signals $s_1$ and $s_2$, and then we can have the ratio masks $M^1 = s_1/S$ and $M^2 = s_2/S$[1]. The goal of the task is to utilize the corresponding visual guidance $v_1$ and $v_2$ to predict the ratio masks for reconstructing the two individual audio signals. This process can be formulated as:

$$
\hat{M}^1 = \mathcal{F}_{\boldsymbol{\Theta}}(\boldsymbol{S}, \boldsymbol{v}_1), \\
\hat{M}^2 = \mathcal{F}_{\boldsymbol{\Theta}}(\boldsymbol{S}, \boldsymbol{v}_2),
\tag{1}
$$

where $\mathcal{F}_{\boldsymbol{\Theta}}$ is the separation model with trainable parameters $\boldsymbol{\Theta}$. And then, the original sound signals $s_1$ and $s_2$ are used to calculate the loss function for optimizing the model:

$$
\boldsymbol{\Theta}^* = \underset{\boldsymbol{\Theta}}{\operatorname{argmin}}\, \mathbb{E}_{(\boldsymbol{V}_1, \boldsymbol{V}_2) \sim \mathcal{D}} \Big[ \mathcal{L}(\hat{\boldsymbol{M}}^1, \boldsymbol{M}^1) + \mathcal{L}(\hat{\boldsymbol{M}}^2, \boldsymbol{M}^2) \Big],
\tag{2}
$$

where $\mathcal{D}$ denotes the training set, and $\mathcal{L}$ is the loss function between the prediction and ground-truth.

**Continual Audio-Visual Sound Separation.** Our proposed continual audio-visual sound separation task aims to train a model $\mathcal{F}_{\boldsymbol{\Theta}}$ continually on a sequence of $T$ separation tasks $\{\mathcal{T}_1, \mathcal{T}_2, ..., \mathcal{T}_T\}$. For the $t$-th task $\mathcal{T}_t$ (incremental step $t$), we have a training set $\mathcal{D}_t = \{\boldsymbol{V}^i(\boldsymbol{s}^i, \boldsymbol{v}^i), y_t^i\}_{i=1}^{n_t}$, where $i$ and $n_t$ denote the $i$-th video sample and the total number of samples in $\mathcal{D}_t$ respectively, and $y_t^i \in \mathcal{C}_t$ is the corresponding sound source class of video $\boldsymbol{V}^i$, where $\mathcal{C}_t$ is the training sound class label space of task $\mathcal{T}_t$. For any two tasks $\mathcal{T}_{t_1}$ and $\mathcal{T}_{t_2}$ and their corresponding training sound class label space $\mathcal{C}_{t_1}$ and $\mathcal{C}_{t_2}$, we have $\mathcal{C}_{t_1} \cap \mathcal{C}_{t_2} = \emptyset$. Following previous works in continual learning [57, 2, 53, 44, 29, 76], for a task $\mathcal{T}_t$, where $t > 1$, holding a small size of memory/exemplar set $\mathcal{M}_t$ to store some data from old tasks is permitted in our setting. Therefore, with the memory/exemplar set $\mathcal{M}_t$, all available data that can be used for training in task $\mathcal{T}_t$ ($t > 1$) can be denoted as $\mathcal{D}'_t = \mathcal{D}_t \cup \mathcal{M}_t$. Finally, the training process of Eq. 2 in our continual audio-visual sound separation setting can be denoted as:

$$
\boldsymbol{\Theta}_t = \underset{\boldsymbol{\Theta}_{t-1}}{\operatorname{argmin}}\, \mathbb{E}_{(\boldsymbol{V}_1, \boldsymbol{V}_2) \sim \mathcal{D}'_t} \Big[ \mathcal{L}(\hat{\boldsymbol{M}}^1, \boldsymbol{M}^1) + \mathcal{L}(\hat{\boldsymbol{M}}^2, \boldsymbol{M}^2) \Big], \\
s.t. \ \ \hat{\boldsymbol{M}}^1 = \mathcal{F}_{\boldsymbol{\Theta}_{t-1}}(\boldsymbol{S}, \boldsymbol{v}_1), \ \hat{\boldsymbol{M}}^2 = \mathcal{F}_{\boldsymbol{\Theta}_{t-1}}(\boldsymbol{S}, \boldsymbol{v}_2),
\tag{3}
$$

which means that the new model $\boldsymbol{\Theta}_t$ is obtained by updating the old model $\boldsymbol{\Theta}_{t-1}$ which was trained on the previous task, using the current task's available data $\mathcal{D}'_t$. After the training process for task $\mathcal{T}_t$ with $\mathcal{D}'_t$, the updated model will be evaluated on a testing set which includes video samples from all seen sound source classes up to continual step $t$ (task $\mathcal{T}_t$). And the evaluation also follows the common "mix-and-separation" strategy. During this continual learning process, the model's separation performance on the previously learned tasks drops significantly after training on new tasks. This learning issue is referred to as the *catastrophic forgetting* [32, 38, 3] problem, which poses a considerable challenge in continual audio-visual sound separation.

## 3.2 Overview

To address the challenge of catastrophic forgetting in continual audio-visual sound separation, we introduce **ContAV-Sep**. This new framework, illustrated in Fig. 2, consists of three key components: a separation base model, an output mask distillation module, and our proposed *Cross-modal Similarity Distillation Constraint (CrossSDC)*. We use a recent state-of-the-art audio-visual separator: iQuery [14] as the base model of our approach, which contains a video encoder to extract the global motion feature, an object detector and image encoder to obtain the object feature, a U-Net [58] for mixture sound encoding and separated sound decoding, and an audio-visual Transformer to get the separated sound feature through multi-modal cross-attention mechanism and class-aware audio queries. For the object detector, we follow iQuery [14] and use the pre-trained Detic [87], a universal object detector, to detect the sound source objects in each frame. For the video encoder and the image encoder, inspired by the excellent generalization ability of recent self-supervised pre-trained models, which has been proven to be effective and appropriate in continual learning [53], we apply two self-supervised pre-trained models VideoMAE [69] and CLIP [56] as the video encoder and the image encoder, respectively. Note that, during the training process, the object detector, video encoder, and image encoder are frozen.

---

[1] In practice, the audio signal is first processed using the Short-Time Fourier Transform (STFT) to generate a spectrogram. For brevity, we will denote spectrogram magnitudes as $s_1$, $s_2$, and $S$.

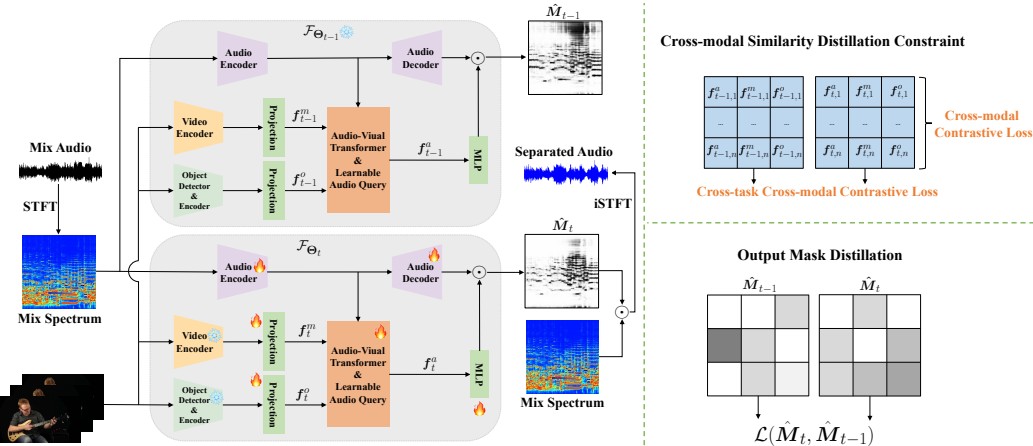

Figure 2: Overview of our proposed ContAV-Sep, which consists of an audio-visual sound separation base model architecture, an Output Mask Distillation, and our proposed Cross-modal Similarity Distillation Constraint. The fire icon denotes the module is trainable, while the snowflake icon denotes that the module is frozen. The (i)STFT stands for (inverse) Short-Time Fourier Transform. Please note that, the old model $\mathcal{F}_{\mathbf{\Theta}_{t-1}}$ is frozen during training.

Given a pair of videos $\boldsymbol{V}_1(\boldsymbol{s}_1, \boldsymbol{v}_1)$ and $\boldsymbol{V}_2(\boldsymbol{s}_2, \boldsymbol{v}_2)$, at incremental step $t$ (task $\mathcal{T}_t$), the U-Net audio encoder $\mathcal{F}_t^{AE}$ takes the mixed audio signal $\boldsymbol{S}$ obtained by mixing $\boldsymbol{s}_1$ and $\boldsymbol{s}_2$ as input, and generates the latent mixed audio feature. This process can be expressed as:

$$\boldsymbol{f}_t^{lat.} = \mathcal{F}_t^{AE}(\boldsymbol{S}), \tag{4}$$

Then, the audio-visual Transformer $\mathcal{F}_t^{Trans.}$ is employed to generate the separated sound feature by taking the latent mixed audio feature and visual features as inputs:

$$\boldsymbol{f}_t^{a,1} = \mathcal{F}_t^{Trans.}(\boldsymbol{f}_t^{lat.}, \boldsymbol{f}_t^{o,1}, \boldsymbol{f}_t^{m,1}),$$
$$s.t. \quad \boldsymbol{f}_t^{o,1} = \boldsymbol{U}_t^o(\boldsymbol{Obj.}^1), \ \boldsymbol{f}_t^{m,1} = \boldsymbol{U}_t^m(\boldsymbol{Mo.}^1), \tag{5}$$

where $\boldsymbol{f}_t^{a,1}$ denotes the separated sound feature of video $\boldsymbol{V}_1$; $\boldsymbol{Obj.}^1$ and $\boldsymbol{Mo.}^1$ denote the object and motion features extracted by the frozen pre-trained image and video encoders respectively from the visual signal $\boldsymbol{v}_1$ of video $\boldsymbol{V}_1$, $\boldsymbol{U}_t^o(\cdot)$ and $\boldsymbol{U}_t^m(\cdot)$ are learnable projection layers to map the object and motion features into the same dimension. Similarly, we can also obtain the separated sound feature of $\boldsymbol{V}_2$ guided by the associated visual features.

The extracted separated sound feature and the latent mixed audio feature are combined to generate a mask. This mask is subsequently applied to the mixed audio, leading to the reconstruction of the separated sound spectrogram.

$$\hat{\boldsymbol{M}}_t^1 = \mathcal{F}_t^{AD}(\boldsymbol{f}_t^{lat.}) \odot MLP_t(\boldsymbol{f}_t^{a,1}),$$
$$\hat{\boldsymbol{M}}_t^2 = \mathcal{F}_t^{AD}(\boldsymbol{f}_t^{lat.}) \odot MLP_t(\boldsymbol{f}_t^{a,2}), \tag{6}$$

where $\hat{\boldsymbol{M}}_t^1$ and $\hat{\boldsymbol{M}}_t^2$ denote the predicted masks for audio signals of video $\boldsymbol{V}_1$ and $\boldsymbol{V}_2$, respectively; $\mathcal{F}_t^{AD}$ is the U-Net decoder at incremental step $t$; $MLP_t(\cdot)$ denotes a MLP module; and $\odot$ denotes channel-wise multiplication. The sound $\boldsymbol{s}_1$ at this incremental step can be reconstructed by applying $\boldsymbol{S} \odot \hat{\boldsymbol{M}}_t^1$ and then performing an inverse STFT to obtain the audio waveform.

## 3.3 Cross-modal Similarity Distillation Constraint

Recent studies [53, 45] have highlighted the importance of cross-modal semantic correlation in audio-visual modeling. However, this correlation tends to diminish during subsequent incremental phases, which leads to catastrophic forgetting in our continual audio-visual sound separation task. To address this challenge, we propose a novel Cross-modal Similarity Distillation Constraint (CrossSDC)

that serves two crucial purposes (1) maintaining cross-modal semantic similarity through incremental tasks, and (2) preserving previous learned semantic similarity knowledge from old tasks.

CrossSDC preserves cross-modal semantic similarity from two perspectives: instance-aware semantic similarity and class-aware semantic similarity. Both similarities are enforced by integrating contrastive loss and knowledge distillation. Instead of exclusively focusing on the similarities within current and memory data generated by the current training model, CrossSDC incorporates the cross-modal similarity knowledge acquired from previous tasks into the contrastive loss. This integration not only facilitates the learning of cross-modal semantic similarities in new tasks but also ensures the preservation of previously acquired knowledge. In the incremental step $t$ ($t > 1$), the instance-aware part of our CrossSDC can be formulated as:

$$\mathcal{L}_{inst.} = -\mathbb{E}_{\boldsymbol{V}^i \sim \mathcal{D}'_t} \left[ \frac{1}{\sum_j \mathbb{1}[i=j]} \sum_j \mathbb{1}[i=j] \log \frac{\exp(\text{sim}(\boldsymbol{f}^{mod_1}_{\tau_1,i}, \boldsymbol{f}^{mod_2}_{\tau_2,j}))}{\sum_k \exp(\text{sim}(\boldsymbol{f}^{mod_1}_{\tau_1,i}, \boldsymbol{f}^{mod_2}_{\tau_2,k}))} \right], \quad (7)$$

where $\mathbb{1}[i=j]$ is an indicator that equals 1 when $i = j$, denoting that video samples $\boldsymbol{V}^i$ and $\boldsymbol{V}^j$ are the same video; The sim function represents the cosine similarity function with temperature scaling; The modalities $mod_1$ and $mod_2$, where $(mod_1, mod_2) \in \{(a,o), (a,m), (m,o)\}$, denote different pairs of features to be compared: separated sound and object features, sound and motion features, and motion and object features. Here, $\tau$ denotes the incremental step, for which we have:

$$\tau_1, \tau_2 \in \mathbf{T}, \; where \; \mathbf{T} = \begin{cases} \{t, t-1\}, & \text{if } \boldsymbol{V} \in \mathcal{M}_t, \\ \{t\}, & \text{if } \boldsymbol{V} \in \mathcal{D}_t, \end{cases} \quad (8)$$

which means that, for current task's data $\mathcal{D}_t$, we calculate the contrastive loss using features from the current model ($\tau_1 = \tau_2 = t$), while for memory set data $\mathcal{M}_t$, we use features from *both the old and current models* (*e.g.,* $\tau_1 = t$ and $\tau_2 = t-1$). In this way, knowledge distillation would be integrated into the cross-modal semantic similarity constraint for the current task, which ensures better preservation of learned cross-modal semantic similarity from previous tasks.

While the instance-aware similarity provides valuable semantic correlation modeling, it does not account for the class-level semantic correlations, which is also crucial for audio-visual similarity modeling. To capture and preserve the semantic similarity within each class across incremental tasks, we also incorporate a class-aware component specifically designed for inter-class cross-modal semantic similarity, which can be formulated as:

$$\mathcal{L}_{cls.} = -\mathbb{E}_{(\boldsymbol{V}^i, y^i) \sim \mathcal{D}'_t} \left[ \frac{1}{\sum_j \mathbb{1}[y^i=y^j]} \sum_j \mathbb{1}[y^i=y^j] \log \frac{\exp(\text{sim}(\boldsymbol{f}^{mod_1}_{\tau_1,i}, \boldsymbol{f}^{mod_2}_{\tau_2,j}))}{\sum_k \exp(\text{sim}(\boldsymbol{f}^{mod_1}_{\tau_1,i}, \boldsymbol{f}^{mod_2}_{\tau_2,k}))} \right]. \quad (9)$$

In this context, visual and audio features from two videos are encouraged to be close when they belong to the same class. The overall formulation of our CrossSDC is as follows:

$$\mathcal{L}_{CrossSDC} = \lambda_{ins}\mathcal{L}_{ins} + \lambda_{cls}\mathcal{L}_{cls}, \quad (10)$$

where $\lambda_{ins}$ and $\lambda_{cls}$ are two scalars that balance the two loss terms. In this way, the model captures and preserves semantic correlations not just between instances but also within the same classes.

## 3.4 Overall Loss Function

In the previous subsection, we introduced our proposed CrossSDC constraint. To effectively combine CrossSDC with the overall objective, we incorporate it alongside output distillation and the main separation loss function.

Output distillation is a widely used technique in continual learning [38, 2, 53] to preserve the knowledge gained from previous tasks while learning new ones. In our approach, we utilize the output of the old model as the distillation target to preserve this knowledge. Note that we only distill knowledge for data from the memory set, as represented by:

$$\mathcal{L}_{dist.} = \mathbb{E}_{(\boldsymbol{V}^i_1, \boldsymbol{V}^i_2) \sim \mathcal{M}_t} \left[ ||\hat{\boldsymbol{M}}^1_t - \hat{\boldsymbol{M}}^1_{t-1}||_1 + ||\hat{\boldsymbol{M}}^2_t - \hat{\boldsymbol{M}}^2_{t-1}||_1 \right], \quad (11)$$

where $\hat{\boldsymbol{M}}^1_{t-1}$ and $\hat{\boldsymbol{M}}^2_{t-1}$ are predicted masks generated by the old model that is trained at incremental step $t-1$. For the loss function here, we follow [84, 14] and adopt the per-pixel $L_1$ loss [84]. For

the main separation loss function, we also apply the per-pixel $L_1$ loss:

$$\mathcal{L}_{main} = \mathbb{E}_{(\boldsymbol{V}_1^i, \boldsymbol{V}_2^i) \sim \mathcal{M}_t} \left[ ||\hat{\boldsymbol{M}}_t^1 - \boldsymbol{M}^1||_1 + ||\hat{\boldsymbol{M}}_t^2 - \boldsymbol{M}^2||_1 \right], \tag{12}$$

Finally, our overall loss function is denoted as:

$$\mathcal{L}_{ContAV-Sep} = \mathcal{L}_{main} + \lambda_{dist.}\mathcal{L}_{dist.} + \mathcal{L}_{CrossSDC}. \tag{13}$$

### 3.5 Management of Memory Set

In alignment with the work of [76], our framework maintains a compact memory set throughout incremental updates. Each old class is limited to a maximum number of exemplars. After completing training for each task, we adopt the exemplar selection strategies in [2, 53] by randomly selecting exemplars for each current class and combining these new exemplars with the existing memory set.

## 4 Experiments

In this section, we first introduce the setup of our experiments, *i.e.*, dataset, baselines, evaluation metrics, and the implementation details. After that, we present the experimental results of our ContAV-Sep compared to the baselines, as well as ablation studies. We also conduct experiments on the AVE [68] and the VGGSound [13] datasets, which contain sound categories beyond the music domain. We put the experimental results on the AVE and the VGGSound datasets, the comparison to the uni-modal semantic similarity preservation method, the performance evaluation on old classes in incremental tasks, and the visualization of separating results in the Appendix.

### 4.1 Experimental Setup

**Dataset.** Following common practice [83, 88, 14], we conducted experiments on *MUSIC-21* [83], which contains solo videos of 21 instruments categories: accordion, acoustic guitar, cello, clarinet, erhu, flute, saxophone, trumpet, tuba, violin, xylophone, bagpipe, banjo, bassoon, congas, drum, electric, bass, guzheng, piano, pipa, and ukulele. In our experiments, we randomly selected 20 of them to construct the continual learning setting. Specifically, we split the selected 20 classes into 4 incremental tasks, each of which involves 5 classes. The total number of available videos is 1040, and we randomly split them into training, validation, and testing sets with 840, 100, and 100 videos, respectively. To further validate the efficacy of our method across a broader sound domain, we conduct experiments using the AVE [68] and the VGGSound [13] datasets in the appendix.

**Baselines.** We compare our proposed approach with vanilla Fine-tuning strategy, and continual learning methods EWC [32] and LwF [38]. As we mentioned before, typical continual learning methods, *e.g.*, class-incremental learning methods, which yield progressively increasing logits (or probability distribution) across all observed classes at each incremental step and design specific technique in the classifier, we consider that these methods are not an optimal choice for our proposed continual audio-visual sound separation problem. Thus, considering that continual semantic segmentation problem has a more similar form compared to conventional class-incremental learning, we also select two state-of-the-art continual semantic segmentation methods PLOP [16] and EWF [78] as our baselines. Moreover, we compare our method to the recently proposed audio-visual continual learning method AV-CIL [53], in which we adapt the original class-incremental version to the form of continual audio-visual sound separation by replacing their task-wise logits distillation with the output mask distillation. Further, we also present the experimental results of the Oracle/Upper Bound, which means that using the training data from all seen classes to train the model. **For fair comparison, all compared continual learning methods and our ContAV-Sep use the same state-of-the-art separator,** *i.e.* **iQuery [14], as the base separation model**. Further, we also incorporate our proposed and baseline methods into another representative audio-visual sound separation model Co-Separation [23]. Notably, the Co-Separation model does not utilize the motion modality. Therefore, when CrossSDC is applied to Co-Separation, the $(mod_1, mod_2)$ in Eq. 7 and 9 is constrained to $(mod_1, mod_2) = (a, o)$. For baselines that involve memory sets, we ensure that each of them is allocated the same number of memory as our proposed method for fair comparison.

**Implementation Details.** Following [14], we use a 7-layers U-Net [58] as the audio net, and subsample the audio at 11kHz, each of which is approximately 6 seconds. We apply the STFT with

Table 1: Main results of different methods on MUSIC-21 dataset under the setting of Continual Audio-Visual Sound Separation with base separation models of iQuery [14] and Co-Separation [23], respectively. The bold part denotes the best results. Our proposed ContAV-Sep achieves the best performance among all baselines.

| Method | SDR↑ | SIR↑ | SAR↑ | Method | SDR↑ | SIR↑ | SAR↑ |
|---|---|---|---|---|---|---|---|
| *w/o memory* | | | | *w/o memory* | | | |
| iQuery [14] + Fine-tuning | 3.46 | 9.30 | 10.57 | Co-Sep. [23] + Fine-tuning | 1.93 | 8.75 | 9.75 |
| iQuery [14] + LwF [38] | 3.45 | 8.78 | 10.66 | Co-Sep. [23] + LwF [38] | 2.32 | 7.84 | 10.28 |
| iQuery [14] + EWC [36] | 3.67 | 9.58 | 10.30 | Co-Sep. [23] + EWC [36] | 2.01 | 8.36 | 9.61 |
| iQuery [14] + PLOP [16] | 3.82 | 10.06 | 10.22 | Co-Sep. [23] + PLOP [16] | 3.24 | 9.17 | 9.59 |
| iQuery [14] + EWF [78] | 3.98 | 9.68 | 11.52 | Co-Sep. [23] + EWF [78] | 2.61 | 7.77 | 10.85 |
| *w/ memory* | | | | *w/ memory* | | | |
| iQuery [14] + LwF [38] | 6.76 | 12.77 | 12.60 | Co-Sep. [23] + LwF [38] | 3.85 | 9.62 | 10.74 |
| iQuery [14] + EWC [36] | 6.65 | 13.01 | 11.73 | Co-Sep. [23] + EWC [36] | 3.31 | 9.55 | 9.80 |
| iQuery [14] + PLOP [16] | 7.03 | 13.30 | 11.90 | Co-Sep. [23] + PLOP [16] | 3.88 | 9.92 | 9.99 |
| iQuery [14] + EWF [78] | 5.35 | 11.35 | 11.81 | Co-Sep. [23] + EWF [78] | 3.63 | 9.07 | 10.58 |
| iQuery [14] + AV-CIL [53] | 6.86 | 13.13 | 12.31 | Co-Sep. [23] + AV-CIL [53] | 3.61 | 9.76 | 9.68 |
| **ContAV-Sep (with iQuery [14])** | **7.33** | **13.55** | **13.01** | **ContAV-Sep (with Co-Sep. [23])** | **4.06** | **10.06** | **11.07** |
| Upper Bound (with iQuery) | 10.36 | 16.64 | 14.68 | Upper Bound (with Co-Sep.) | 7.30 | 14.34 | 11.90 |

the Hann window size of 1022 and the hop length of 256, to obtain the $512 \times 256$ Time-Frequency representation of each audio signal, followed by a re-sampling on the log-frequency scale to generate the magnitude spectrogram with $T, F = 256$. We set the video frame rate (FPS) to 1, and detect the object using the pre-trained universal detector Detic [87] to detect the sound source object on each frame, and then, each detected object is resized and randomly cropped to the size of $224 \times 224$. For the image encoder and the video encoder, we apply the self-supervised pre-trained CLIP [56] and VideoMAE [69] to yield the object feature and motion feature, respectively. For the audio-visual Transformer module, we follow the design in [14]. For all the baseline methods, we apply the same model architecture and modules with ours for them, including the mentioned Detic, CLIP, VideoMAE, audio-visual Transformer, etc. Please note that, during our training process, the pre-trained Detic, CLIP, and VideoMAE are frozen. In our proposed Cross-modal Similarity Distillation Constraint (CrossSDC), the balance weights $\lambda_{ins}$ and $\lambda_{cls}$ are set to 0.1 and 0.3, respectively. And the balance weight $\lambda_{dist.}$ for the output distillation loss is set to 0.3 in our experiments. For the memory set, we set the number of samples in each old class to 1, so as other baselines that involve the memory set. All the experiments in this paper are implemented by Pytorch [51]. We train our proposed method and all baselines on a NVIDIA RTX A5000 GPU. We follow previous works [67, 14] in sound separation, and evaluate the performance of all the methods using three common metrics in sound separation tasks: Signal to Distortion Ratio (SDR), Signal to Interference Ratio (SIR), and Signal to Artifact Ratio (SAR). The SDR measures the interference and artifacts, while SIR and SAR measure the interference and artifacts, respectively. In our experiments, we report the SDR, SIR, and SAR of all the methods after training at last incremental steps, *i.e.*, testing results on all classes. For all these three metrics, higher values denote better results.

## 4.2 Experimental Comparison

The main experimental comparisons are shown in Tab. 1. Our proposed method, ContAV-Sep, outperforms the state-of-the-art baselines by a substantial margin. Notably, compared to baselines using state-of-the-art audio-visual sound separator iQuery [14] as the separation base model, ContAV-Sep achieves a 0.3 improvement in SDR over the compared best-performing method. Additionally, our method surpasses the top baseline by 0.25 in SIR and 0.41 in SAR. Furthermore, compared to continual learning baselines with Co-Separation [23], our ContAV-Sep still outperforms other approaches. This consistent superior performance across different model architectures highlights not only the effectiveness but also the broad applicability and generalizability of our proposed CrossSDC.

Our observations further demonstrate that retaining a small memory set significantly enhances the performance of each baseline method. For instance, for the iQuery-based continual learning methods, equipping LwF [38] with a small memory set results in improvements of 3.31, 3.99, and 1.94 on SDR, SIR, and SAR, respectively. Similarly, the addition of a small memory set to EWC [32] leads to enhancements of 2.98, 3.43, and 1.43 in the respective metrics. The memory-augmented version of PLOP [16] exhibits superior performance with margins of 3.21, 3.24, and 1.68 for SDR, SIR, and SAR, respectively. Finally, incorporating memory into EWF [78] results in improvements of 1.37,

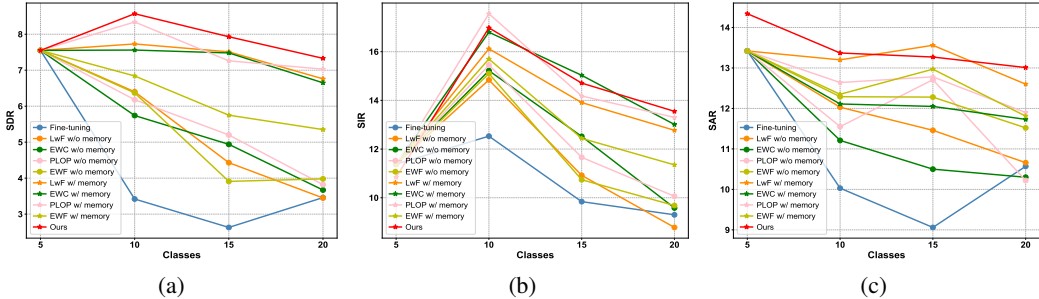

(a)          (b)          (c)

Figure 3: Testing results of different continual learning methods with iQuery [14] on the metrics of (a) SDR, (b) SIR, and (c) SAR at each incremental step.

1.67, and 0.29 for the three metrics. This phenomenon can be attributed to the inherent nature of the sound separation training process. In training, the audio signal from each sample mixes with others, giving a composite audio signal. This mixed audio signal, coupled with the corresponding visual data pair for each separated audio, constitutes the actual training sample for the separation task. As a result, even a single memory sample can be associated with multiple samples from the current training set, generating a diverse array of effective training pairs.

We also present the testing results of SDR, SIR, and SAR at each incremental step in Figures 3a, 3b, and 3c, respectively. Our method is consistently observed to outperform others in terms of SDR at all incremental steps. While our approach may not always produce the best SIR and SAR results at the intermediate steps (specifically, steps 2 and 3 for SIR, and step 3 for SAR), it ultimately achieves the highest performance at the final step. This demonstrates the robustness of our method, indicating minimal forgetting throughout the incremental learning process.

## 4.3 Ablation Study on CrossSDC and Memory Size

In this subsection, we conduct an ablation study to investigate the effectiveness of our proposed CrossSDC. By removing single or multiple components of the CrossSDC, we evaluate the impact of each on the final results. The results of the ablation study are presented in Tab. 2. From the table, we can see that our full model achieves the best performance compared to the variants, which further demonstrates the effectiveness of our proposed CrossSDC.

Moreover, we also discuss the effect of memory size on our proposed ContAV-Sep. In our main experiments, the default setting of the memory size is 1 sample per old class. In this subsection, we conduct experiments by increasing the memory size from 1 sample per old class to 30 samples per old class. The experimental results are shown in Tab. 3 and Figure 4. Observations from the table indicate a positive correlation between the size of the memory and the overall performance metrics. As the memory size increases, there is a discernible trend of improvement in the results.

Table 2: Ablation study on our proposed ContAV-Sep. Our full approach achieves best results compared to the variants.

|  | $\mathcal{L}_{dist.}$ | $\mathcal{L}_{inst.}$ | $\mathcal{L}_{cls.}$ | SDR↑ | SIR↑ | SAR↑ |
|---|---|---|---|---|---|---|
| ContAV-Sep | ✔ | ✘ | ✘ | 6.32 | 12.99 | 11.82 |
|  | ✔ | ✔ | ✘ | 6.01 | 11.92 | 11.74 |
|  | ✔ | ✘ | ✔ | 6.86 | 13.12 | 12.25 |
|  | ✔ | ✔ | ✔ | **7.33** | **13.55** | **13.01** |

## 4.4 Limitation and Discussion

Our experimental findings reveal that the utilization of a small memory set, even a single sample per old class, markedly improves the performance of each baseline method. This improvement is attributed to the ability of a single memory sample to pair with diverse samples from the current training set, thereby generating numerous effective training pairs. Consequently, this process enables the model to acquire new knowledge for old classes in subsequent tasks, as the memory data can be

Table 3: Experimental results of our proposed ContAV-Sep with different memory size from 1 to 30 samples per memory class.

|  | # of samples per class | SDR↑ | SIR↑ | SAR↑ |
|---|---|---|---|---|
| ContAV-Sep | 1 | 7.33 | 13.55 | 13.01 |
|  | 2 | 7.26 | 13.10 | 12.65 |
|  | 3 | 7.88 | 13.66 | 13.43 |
|  | 4 | 8.16 | 14.16 | 13.21 |
|  | 10 | 8.97 | 15.16 | 13.72 |
|  | 20 | 9.39 | 15.93 | 13.69 |
|  | 30 | **10.09** | **16.34** | **14.10** |

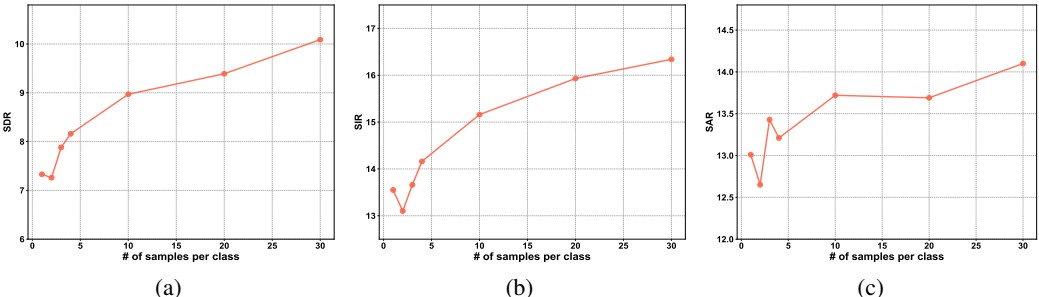

(a)          (b)          (c)

Figure 4: Testing results with different memory size (number of samples per class in the memory) on the metrics of (a) SDR, (b) SIR, and (c) SAR at each incremental step.

paired with data from previously unseen new classes — this is different from conventional continual learning tasks, where old classes do *not* acquire new knowledge in new tasks. This could be a potential reason why the baseline continual learning methods do not perform well in our continual audio-visual sound separation problem. In this work, our method also mainly focuses on preserving old knowledge of old tasks, which may prevent the model from acquiring new knowledge of old classes when training in new tasks. Recognizing this, we identify the exploration of this problem as a key avenue for future research in this field.

Additionally, the base model architectures used in our approach and baselines require object detectors to identify sounding objects. Although iQuery [14] can supplement object features with global video representations, it may still suffer from undetected objects. It is a fundamental limitation of the **object-based audio-visual sound separators** [23, 14]. While our work, unlike previous efforts, does not compete on designing a stronger audio-visual separation base model, enhancing the robustness of sounding object detection presents a promising direction for future research.

## 5 Conclusion

In this paper, we explore training audio-visual sound separation models under a more practical continual learning scenario, and introduce the task of continual audio-visual sound separation. To address this novel problem, we propose ContAV-Sep, which incorporates a Cross-modal Similarity Distillation Constraint to maintain cross-modal semantic similarity across incremental tasks while preserving previously learned semantic similarity knowledge. Experiments on the MUSIC-21 dataset demonstrate the effectiveness of our method in this new continual separation task. This paper opens a new direction for real-world audio-visual sound separation research.

**Broader Impact.** Our proposed continual audio-visual sound separation allows the model to adapt to new environments and sounds without full retraining, which could enhance efficiency and privacy by reducing the need to transmit and store sensitive audio data.

**Acknowledgments.** We thank the anonymous reviewers and area chair for their valuable suggestions and comments. This work was supported in part by a Cisco Faculty Research Award, an Amazon Research Award, and a research gift from Adobe. The article solely reflects the opinions and conclusions of its authors but not the funding agents.

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
