# OpenReview forum: "Continual Audio-Visual Sound Separation"
_NeurIPS.cc/2024/Conference — NeurIPS 2024 poster_

### Official Review · Reviewer_tY1x · 2024-07-08

**Soundness:** 2
**Presentation:** 3
**Contribution:** 1
**Rating:** 5
**Confidence:** 4

**Summary:**

The paper proposes a novel continual audio-visual sound separation task, aimed at continuously separating new categories of sound sources while maintaining the performance of previously learned categories.

**Strengths:**

1. The structure of the entire paper is clear, and the expression is fluent.
2. The experimental results can demonstrate the effectiveness of the method, and the quality of the separated audio shown in the visual video is also good.

**Weaknesses:**

1. The Cross-modal Similarity Distillation Constraint proposed in the paper includes two main innovations: 1) instance-aware semantic similarity and 2) class-aware semantic similarity. However, in terms of mathematical expression, compared to the Dual-Audio-Visual Similarity Constraint in reference [1], I believe the innovations of the two papers are almost identical.
[1] Pian W, Mo S, Guo Y, et al. Audio-visual class-incremental learning[C]//Proceedings of the IEEE/CVF International Conference on Computer Vision. 2023: 7799-7811.
2. Although the paper has conducted experimental validation on the MUSIC-21 dataset, it does not provide sufficient information to assess the model's generalization ability on other datasets or in real-world scenarios, making it difficult to demonstrate its enhanced applicability in real scenarios where new sound sources are encountered.

**Questions:**

1. In Fig. 1, the authors illustrate the continual audio-visual sound separation task. Is the significant performance gap between ContAV-Sep and the fine-tuning method due to the small data scale and limited number of categories?

2. I am curious whether this paper has been submitted twice, as I noticed the "Anonymous ECCV Submission Paper ID 6024" in the uploaded video materials. If that is the case, this paper will be directly rejected.

3. Please clarify the difference between the innovation of the Cross-modal Similarity Distillation Constraint in this paper and the Dual-Audio-Visual Similarity Constraint in [1].

**Limitations:**

The innovation of the Cross-modal Similarity Distillation Constraint in this paper may overlap with the Dual-Audio-Visual Similarity Constraint from the previous work. It suggests that while the paper introduces a novel approach, there might not be a clear distinction or significant difference from the existing method in terms of the core innovative aspects.

---

> ### Author Rebuttal · Authors · 2024-08-05
>
> Thank you for the valuable comments! We appreciate the reviewer highlighting the clear structure, fluent expression, and effective experimental results of our paper. We address the raised concerns below and are willing to answer any further questions.
>
> > ### **Q1: Difference between the innovation of the Cross-modal Similarity Distillation Constraint (CrossSDC) and Dual-Audio-Visual Similarity Constraint (D-AVSC) in [1].**
>
> Thank you for your question! The Dual-Audio-Visual Similarity Constraint (D-AVSC) [1] only focuses on the feature similarities within data generated by the current training model, while our proposed CrossSDC further integrate features learned by previous model into the contrastive loss. In this way, our CrossSDC incorporates the cross-modal similarities knowledge acquired from previous tasks into the contrastive loss, which not only facilitates the learning of cross-modal semantic similarities in new tasks but also ensures the preservation of previously acquired knowledge in audio-visual sound separation.
>
> Moreover, the experimental results also demonstrate the superiority of our method with CrossSDC (**7.33/13.55/13.01** for SDR/SIR/SAR) compared to the AV-CIL with D-AVSC [1] (6.86/13.13/12.31 for SDR/SIR/SAR).
>
>
> > ### **Q2: Generalization ability on other datasets.**
>
> Thanks for the suggestion! Besides the MUSIC-21 dataset, we conducted experiments on the AVE [2] dataset and have included the experimental results in Appendix. The results of Fine-tuning/LwF/PLOP/AV-CIL/ContAV-Sep(Ours) are 2.07/2.19/2.45/2.53/**2.72** and 5.64/6.43/6.11/6.64/**7.32** for SDR and SIR, respectively, which further demonstrates the effectiveness of our proposed method. For more details, please kindly see Section A.3 in Appendix.
>
>
> > ### **Q3: Is the significant performance gap between ContAV-Sep and the fine-tuning method due to the small data scale and limited number of categories?**
>
> We would like to clarify that the significant performance gap between ContAV-Sep and fine-tuning is not due to the small data scale and limited number of categories. Experiments on a larger dataset, AVE [2], demonstrate a similar performance gap. On AVE, fine-tuning achieves an SDR of 2.07 and SIR of 5.64, while ContAV-Sep achieves **2.72** and **7.32**, respectively. This persistent gap underscores that the difference is not solely due to the dataset. For further details, please refer to Section A.3 in the Appendix.
>
>
> > ### **Q4: Whether this paper has been submitted twice?**
>
>  We confirm that this paper is not a dual submission. The current version is the revised submission of our previously withdrawn paper. We apologize for the oversight in the teaser of the demo video, which contained a typo. This issue has been corrected in our demo.
>
>
>
> [1] Audio-Visual Class-Incremental Learning. In *ICCV* 2023.
>
> [2] Audio-Visual Event Localization in Unconstrained Videos. In *ECCV* 2018.

---

> > ### Comment · Reviewer_tY1x · 2024-08-11
> >
> > Thank for authors' response. However, the technique novelty is limited. The proposed CrossSDC just integrates features learned by previous model into the contrastive loss upon the D-AVSC,  so I hold on my score without any modification.

---

> > > ### Author Response · Authors · 2024-08-11
> > > **Official Comment by Authors**
> > >
> > > Thank you, Reviewer tY1x, for your response! We respect your opinion but respectfully disagree that the technical novelty is limited. In this work, we introduce not only the novel task of Continual Audio-Visual Sound Separation but also CrossSDC to address it. We’d like to emphasize that, in addition to the model, the new task itself is an important contribution to the field.
> > >
> > > Furthermore, we believe our CrossSDC is indeed a novel approach within continual audio-visual sound separation. By integrating features from previous tasks into the cross-modal contrastive loss, we are pioneering the maintenance of audio-visual similarity across tasks. This allows us to learn audio-visual similarity within the current task and distill similarity from previous tasks, all without requiring a specific model architecture designed to combat catastrophic forgetting. Our approach is simple yet effective, and experiments with two different separators on two datasets demonstrate its superiority.
> > >
> > > We respectfully leave the final judgment to you, the other reviewers, and the AC. We sincerely appreciate your comments and engagement with our rebuttal!

---

> > > > ### Comment · Reviewer_tY1x · 2024-08-12
> > > >
> > > > Thank you for the author's further response and the keen interest in delving deeper into the reasons. As the author mentioned, the paper introduces a new task, 'Continual Audio-Visual Sound Separation,' by combining continual learning with audio-visual sound separation. However, this approach originates from the combination of continual learning and audio-visual classification tasks, making it easily extendable. For instance, one could combine continual learning with audio-visual segmentation or audio-visual event localization, which makes proposing this task relatively straightforward. As for CrossSDC, most of its content is derived from Audio-Visual Class-Incremental Learning, which is not a significant innovation.
> > > >
> > > > **My core question is: how significant is the value of combining two existing tasks to create a new one? Can extending the innovative aspects from previous similar methods to this new task be considered a major innovation?**

---

> > > > > ### Author Response · Authors · 2024-08-12
> > > > > **Response to Reviewer tY1x**
> > > > >
> > > > > Thank you for your insightful questions! We want to clarify that solving audio-visual sound separation in a continual learning setting is non-trivial. Extending continual learning from classification to this sound separation is far from straightforward due to fundamental differences between the two.
> > > > >
> > > > > In continual audio-visual classification, the goal is to accurately identify the category of an audio-visual clip by extracting and integrating discriminative features across tasks, while avoiding catastrophic forgetting. In contrast, audio-visual sound separation is inherently more complex. It requires using visual signals to disentangle mixed sounds into visually-aligned audio features, ultimately predicting ratio sound spectrogram masks for separating individual sources. Here, learning discriminative features alone is insufficient. Instead, preserving the relative correlations or similarities between audio and visual components is more critical and difficult in the separation task, as various sound sources can be mixed with a source from a known or unknown class.
> > > > >
> > > > > Therefore, we consider Continual Audio-Visual Sound Separation a more challenging task than continual audio-visual classification. The core challenge lies in maintaining these relative audio-visual correlations or similarities to leverage visually indicated sound representations for separation, all while mitigating forgetting. This ensures the preservation of separation abilities learned from previous classes. Otherwise, catastrophic forgetting affects not only old classes but also new class separation, particularly when new sound is mixed with data from previously learned classes.
> > > > >
> > > > > Moreover, in continual classification, the model is trained to predict continually incrementing logits among all previously seen classes. However, our proposed Continual Audio-Visual Sound Separation task focuses on addressing catastrophic forgetting in the context of separation mask prediction within joint audio-visual modeling. Unlike in continual classification, where the model adapts to an expanding output space as new classes are introduced, our task requires the model to consistently predict a fixed-size separation mask with values between 0 and 1 across all tasks. This presents a greater challenge because the model must retain its ability to effectively separate audio sources even as new tasks are introduced, without expanding the output space. Consequently, directly applying continual learning methods from classification to our proposed task is not feasible and requires significant adaptation, especially for methods involving techniques focused on partially updating the classifier for new classes, as evidenced by our experiments and demo.
> > > > >
> > > > > Our CrossSDC, unlike D-AVSC in [1], goes beyond focusing solely on feature similarities within the current task’s training model. It integrates previously learned features into the contrastive learning process, establishing a cross-task audio-visual correlation/similarity constraint. This enables the preservation and maintenance of relative audio-visual correlation/similarities across current and all seen tasks. Our experiments demonstrate that CrossSDC significantly outperforms AV-CIL with D-AVSC on various datasets and with different separation architectures.
> > > > >
> > > > > [1] Audio-Visual Class-Incremental Learning. In ICCV 2023.

---

> > > > > > ### Comment · Reviewer_tY1x · 2024-08-13
> > > > > >
> > > > > > Based on the author's answer, although it elaborates a lot on the differences between continuous audio-visual separation and continuous audio-visual classification, it feels partially answering the question: **how significant is the value of combining two existing tasks to create a new one**, as tasks such as Continual Audio-Visual Segmentation may come up later, and they may all be similar in approach to this paper, making it it is difficult to assess their impact. But I've decided to raise my rating as you are actively working on these issues.

---

> > > > > > > ### Author Response · Authors · 2024-08-13
> > > > > > > **Response to reviewer tY1x**
> > > > > > >
> > > > > > > Thank you for raising your rating! We sincerely appreciate your insightful questions and feedback, and we're grateful for your patience in reviewing our detailed responses.
> > > > > > >
> > > > > > > You've raised an excellent point. Our approach could potentially be extended to address other audio-visual prediction tasks such as sound source visual localization and audio-visual segmentation. However, these tasks present additional challenges, as they require learning very fine-grained correspondences between individual sound sources and their corresponding visual objects in order to accurately predict the visual regions associated with specific sounds in video frames. Furthermore, these tasks may only have access to weak supervision in the form of video-level labels during training, while needing to predict precise sounding regions or segmentation masks in testing. This would necessitate the development of novel multimodal continual learning approaches, opening up new technical challenges and exciting avenues for future research.

---

### Official Review · Reviewer_wmVP · 2024-07-10

**Soundness:** 3
**Presentation:** 3
**Contribution:** 3
**Rating:** 6
**Confidence:** 4

**Summary:**

This paper introduces a novel task termed "Continual Audio-Visual Sound Separation," aiming to address the practical challenge of separating sound sources for new classes in audio-visual scenarios while retaining performance on previously learned classes. This task is inherently challenging due to the inherent risk of catastrophic forgetting, where models trained on new data often exhibit performance degradation on previously learned classes. To tackle this challenge, the authors propose ContAV-Sep, a novel framework incorporating a Cross-modal Similarity Distillation Constraint (CrossSDC). This constraint preserves cross-modal semantic similarity across incremental tasks by enforcing both instance-aware and class-aware similarities through a combination of contrastive loss and knowledge distillation. Notably, CrossSDC integrates knowledge from past tasks into the contrastive learning process, ensuring the retention of previously acquired cross-modal correlations. Experiments on the MUSIC-21 dataset demonstrate that ContAV-Sep significantly outperforms existing continual learning baselines in terms of standard sound separation metrics (SDR, SIR, SAR) across multiple audio-visual sound separation base models. This work highlights the importance of cross-modal semantic correlation in continual audio-visual learning and provides a novel, effective solution for mitigating catastrophic forgetting in this domain.

**Strengths:**

Here are four strengths of the paper, presented as a list:
- Clearly identifies and addresses a novel and practical problem in audio-visual sound separation: continual learning in this domain. It is important to mention that this might be an important problem in the field and not so many people have provided good solutions for this problem.
- Proposes a novel framework, ContAV-Sep,  with a well-defined Cross-modal Similarity Distillation Constraint (CrossSDC) to tackle catastrophic forgetting.
- Empirically demonstrates the effectiveness of ContAV-Sep on the MUSIC-21 dataset, showcasing significant performance improvements over strong baselines.
- Provides a thorough analysis of the results, including ablation studies and exploration of memory size effects, highlighting the contributions of different components of the proposed method.

**Weaknesses:**

- Please replace the “mask” variable with something appropriate like \widehat{\mathbf{m}}
- Please fix weird fontsizes like the one in equation 9, in general the manuscript does not seem polished enough for a NeurIPS submission.
- Differences smaller than <0.1 dB in terms of SNR metrics are not either significant nor hearable (I would even argue for 0.5 dB but let’s follow the literature in this one), thus I would suggest rounding up all those performance numbers to a one decimal precision (it would also make the Tables less cluttered).
- I would like to see an even larger ablation in Table 3 to show the full extent of how the performance deviates with an even larger amount of samples per class and not only 4, a graph would make the visualization better here cause currently it does not convey any meaningful message.
- The authors do not make a thorough investigation on previous works in the literature that employ continual learning techniques for related sound processing tasks. I will refer here only a couple of the works that I am aware of like in [A, B] but I am almost certain that there is no lack of thereof to try to include other works and try to make some empirical or theoretical conclusions on how those methods can become interconnected, employed together and in general how they relate.



[A] Wang, Z., Subakan, C., Tzinis, E., Smaragdis, P. and Charlin, L., 2019, October. Continual learning of new sound classes using generative replay. In 2019 IEEE Workshop on Applications of Signal Processing to Audio and Acoustics (WASPAA) (pp. 308-312). IEEE.

[B] Wang, Z., Subakan, C., Jiang, X., Wu, J., Tzinis, E., Ravanelli, M. and Smaragdis, P., 2022. Learning representations for new sound classes with continual self-supervised learning. IEEE Signal Processing Letters, 29, pp.2607-2611.

**Questions:**

The classes seem a bit limited, would the authors want to generalize their experiments to a larger dataset like audioset and show how their method performs there? I think that would be even more interesting since the distribution of classes in Audioset has a long tail for multiple not so prevalent sounds (e.g. consider sounds that are not represented by Speech or music)

**Limitations:**

I think the authors include several limitations of their work, they should also identify the memorization of individual user’s data inside the memory of their continual learning method.

---

> ### Author Rebuttal · Authors · 2024-08-05
>
> Thank you for your valuable comments and suggestions! We appreciate your recognition of the novelty in our work's problem formulation and approach, as well as its effectiveness demonstrated in our experiments. We address your questions below and welcome any further inquiries.
>
> > ### **Q1: Replace the "**mask**" variable.**
>
> Thank you for your suggestion. We have replaced the "**mask**" variable with a variable **m**.
>
> > ### **Q2: Weird fontsizes like the one in equation 9.**
>
> Thank you for your suggestion. We have fixed this fontsize issue.
>
> > ### **Q3: Rounding up all performance numbers to a one decimal precision.**
>
> Nice suggestion. We have updated performance numbers to one decimal precision in our paper.
>
> > ### **Q4: Larger amount of samples per class in memory. A graph would make the visualization better.**
>
> Thank you for your suggestion! We conducted experiments with the number of samples per class in memory with 10, 20, and 30. The results are shown below. Moreover, as per your advice, we have included a graph in our paper to better show the impact of different sample sizes per class in memory.
>
>
> | # of samples per class | SDR  | SIR  | SAR  |
> |------------------------|------|------|------|
> | 10                     | 9.0  | 15.2 | 13.7 |
> | 20                     | 9.4  | 15.9 | 13.7 |
> | 30                     | 10.1 | 16.6 | 14.1 |
>
>
>
> > ### **Q5: Missing references in continual learning for sound processing tasks.**
>
> Thanks for the suggestion! We have added relevant references on continual learning for sound processing tasks, including the two suggested works and others [1, 2, 3, 4], and discussed them in our revised paper:
>
> [1] Ma, Haoxin, Jiangyan Yi, Jianhua Tao, Ye Bai, Zhengkun Tian, and Chenglong Wang. "Continual learning for fake audio detection." arXiv preprint arXiv:2104.07286 (2021).
>
> [2] Bhatt, Ruchi, Pratibha Kumari, Dwarikanath Mahapatra, Abdulmotaleb El Saddik, and Mukesh Saini. "Characterizing Continual Learning Scenarios and Strategies for Audio Analysis." arXiv preprint arXiv:2407.00465 (2024).
>
> [3] Wang, Yu, Nicholas J. Bryan, Mark Cartwright, Juan Pablo Bello, and Justin Salamon. "Few-shot continual learning for audio classification." In ICASSP 2021-2021 IEEE International Conference on Acoustics, Speech and Signal Processing (ICASSP), pp. 321-325. IEEE, 2021.
>
> [4] Zhang, Xiaohui, Jiangyan Yi, Jianhua Tao, Chenglong Wang, and Chu Yuan Zhang. "Do you remember? overcoming catastrophic forgetting for fake audio detection." In International Conference on Machine Learning, pp. 41819-41831. PMLR, 2023.
>
> > ### **Q6: Generalize the experiments to a larger dataset like AudioSet.**
>
> Nice suggestion! Following common practice, we used the MUSIC dataset as our primary benchmark and included an additional dataset, AVE, in the appendix to validate audio-visual sound separation performance. Extending our approach to very large datasets like AudioSet would indeed be interesting. However, due to limited computational resources and time constraints, conducting extensive experiments on AudioSet is currently challenging. We will investigate this in our future work.
>
> > ### **Q7: Identify the memorization of individual user’s data inside the memory.**
>
> Thank you for your suggestion. We will address the potential memorization of individual users' data within our continual learning method in the limitations section. This discussion will include the importance of future work to empirically analyze and prevent data memorization, ensuring the development of robust and privacy-preserving models.

---

> > ### Author Response · Authors · 2024-08-12
> > **Official Comment by Authors**
> >
> > Dear Reviewer wmVP,
> >
> > We sincerely appreciate your valuable feedback, which has greatly improved our paper! We kindly ask if you could confirm whether our response has adequately addressed your concerns. If so, we would be grateful if you might consider raising your rating. Please do not hesitate to let us know if there are any remaining issues.
> >
> > Thank you once again for your insightful feedback!
> >
> > Best Regards,
> >
> > The Authors

---

> > > ### Author Response · Authors · 2024-08-12
> > > **Experiments on larger dataset with broader range of sound categories**
> > >
> > > Dear Reviewer wmVP,
> > >
> > > To demonstrate that our approach can generalize to larger dataset with broader range of sound categories, we have conducted experiments on the VGGSound dataset, in which we use randomly selected 100 categories for our continual audio-visual sound separation. We train both our method and the baseline methods using iQuery as the separator architecture. The results for Fine-tuning/LwF/PLOP/AV-CIL/ContAV-Sep (Ours) are 3.69/4.71/4.56/4.66/**4.90** for SDR and 7.23/8.89/8.32/8.61/**9.25** for SIR, respectively, which further demonstrate the effectiveness of our approach on a broader range of data categories.
> > >
> > > Best Regards,
> > >
> > > The Authors

---

> > > > ### Comment · Reviewer_wmVP · 2024-08-12
> > > > **Response to the authors' rebuttal**
> > > >
> > > > Thanks for addressing some of my concerns and my questions. I congratulate the authors' rebuttal and the extra experiments especially for the larger dataset of VGGsound. I will thus, increase my score, although the comparisons with other methods in the literature are still not really easy to me how they used same model capacity etc.

---

> > > > > ### Author Response · Authors · 2024-08-12
> > > > > **Response to reviewer wmVP**
> > > > >
> > > > > Thank you so much for your positive support of our work! Regarding your question about how different methods use the same architecture, we would like to clarify that continual learning methods typically involve different strategies for training on new tasks, such as selective weight updating, memory buffer usage, or the incorporation of regularization techniques to prevent catastrophic forgetting. As a result, most continual learning methods can be viewed as specific training strategies that are independent of the model architecture. Therefore, a common experimental setting in continual learning research is to hold the model architecture constant across the proposed method and all baseline methods, allowing for a fair comparison of their training performance.
> > > > >
> > > > > We will provide more details about the implementation details in appendix and release our source code for our model and the implemented baselines. We truly appreciate your support!

---

### Official Review · Reviewer_5v9b · 2024-07-12

**Soundness:** 3
**Presentation:** 3
**Contribution:** 3
**Rating:** 5
**Confidence:** 3

**Summary:**

The paper introduces ContAV-Sep, the goal is to continuously separate new sound classes while maintaining performance on previously learned classes, addressing the challenge of catastrophic forgetting in continual learning. ContAV-Sep employs a Cross-modal Similarity Distillation Constraint (CrossSDC) to preserve cross-modal semantic similarity across tasks and retain old knowledge, integrated into an audio-visual sound separation framework.

**Strengths:**

1. The proposed method is a good solution to help maintain cross-modal semantic similarity across incremental tasks.
2. The paper has a clear writing structure, and the figures and tables are easy to understand.

**Weaknesses:**

1. The performance improvement of ContAV-Sep (with iQuery) is not significant.
2. What does iSTFT stand for in Figure 2? The symbols appearing in the figure need to be clearly explained in the caption.

**Questions:**

See weakness.

**Limitations:**

See weakness.

---

> ### Author Rebuttal · Authors · 2024-08-05
>
> We appreciate the reviewer highlighting our proposed approach and writing. We address the raised questions below and are happy to answer further questions.
>
> > ### **Q1: The performance improvement of ContAV-Sep (with iQuery) is not significant.**
>
> Thanks for the comment! We would like to clarify that the performance improvement of our ContAV-Sep (with iQuery) is non-trivial. Compared to the baseline continual learning methods LwF/EWC/PLOP/EWF/AV-CIL, our proposed ContAV-Sep shows improvements of 0.57/0.68/0.30/1.98/0.47 in SDR and 0.78/0.54/0.25/2.20/0.42 in SIR, respectively. Furthermore, as demonstrated in the demo included in our supplementary material, our ContAV-Sep (with iQuery) achieves better separation results and sound quality compared to the baselines, highlighting its effectiveness in mitigating catastrophic forgetting.
>
>
> > ### **Q2: What does iSTFT stand for in Figure 2?**
>
> Thank you for your question. iSTFT stands for Inverse Short-Time Fourier Transform, a common technique used to convert spectrograms back into audio signals. We have added this explanation to the figure caption for clarity.

---

> > ### Author Response · Authors · 2024-08-12
> > **Official Comment by Authors**
> >
> > Dear Reviewer 5v9b,
> >
> > We sincerely appreciate your valuable feedback, which has greatly improved our paper! We kindly ask if you could confirm whether our response has adequately addressed your concerns. If so, we would be grateful if you might consider raising your rating. Please do not hesitate to let us know if there are any remaining issues.
> >
> > Thank you once again for your insightful feedback!
> >
> > Best Regards,
> >
> > The Authors

---

### Official Review · Reviewer_49mS · 2024-07-14

**Soundness:** 3
**Presentation:** 4
**Contribution:** 3
**Rating:** 7
**Confidence:** 4

**Summary:**

This approach defines an audio-visual sound separation task where sound separation is the task and during fine-tuning novel classes are added, in the regime of continuous learning. The goal is to avoid catastrophic forgetting which typically leads to decreased performance in task performance on classes which were learned early but then little or no instances arrive in later training stages. They then present a method to solve this task with audio-visual data. Several losses are defined to achieve cross-modal similarity of embeddings through incremental tasks and preserve previous learned knowledge.

Their approach does not increase the number of logits to be able to accommodate more classes but instead creates separation masks, more suitable to audio separation than typical classification tasks.

Results are compared on meaningful baselines even though this is basically a new task definition and ablating the different loss components shows that the combination yields the best performance.

Additional videos and investigations in the supplementary work complete the work.

**Strengths:**

Especially in the videos the performance on very similar instruments is impressive. The instruments share a very similar frequency profile and their separation is a very hard problem.

The paper is very clearly written, the task is clearly stated and well contrasted and localized within the domain.

The mathematical notation is very clear to follow.

In general the paper defines an interesting and realistic task, does a thorough investigation and states clearly its limitations.

**Weaknesses:**

The paper defines a task and then solves it, which is always a bit easier due to limited competition. A direct comparison between sound separation was excluded by the authors stating that they do not compete in that sense, which is a limitation.

Figure 2 is very small and could be more self complete. The relation of the right side's illustration to the left side is not clear from the caption at all which only states the names of the concepts. Also, there is space left and right to increase the size. Probably this resizing was reduced to gain space. This leaves the text very small. It would be good to find space in a different way because the fonts are already extremely small within Figure 2. Some of the dataset details could probably go into the appendix.

This is not a summary paper. That the author's found so much related work is commendable but I would have preferred more detailed contrast to existing work, e.g. by picking out certain ideas and contributions, explaining them and then contrasting this work against them. Just having lists of 7 works and summarizing them as "semantic segmentation" seems more like a homework chore than a contribution to the paper. But my fellow reviewers may disagree.

**Questions:**

- It seems from the problem definition that all tasks have distinct classes. That seems like a special, even though the hardest possible, case. Do you have an intuition or looked into variations where actually a smaller part of instances from a specific class is part of all tasks? It seems also more realistic to assume that such a network is not necessarily fine-tuned on a dataset that is completely different from the previous one.

- What are the hardware requirements for this? It seems CLIP is used even twice, once for each task and then the features are fed into transformers. Yet the NVIDIA RTC A5000 does seem to have only 24GB. Are there some compromises, e.g. image resolution, which were made due to the hardware available?

- On a related note: What is the memory size? How does the memory and computation scale to new tasks as the memory keeps growing beyond 4 tasks?

- The choice of 11kHz audio sampling seems odd compared with defaults in audio encoding or other approaches for other audio-visual tasks. What is the reasoning behind it?

- How does this scale to hard examples, where both instruments are the same. The second example is already impressive with two woodwinds but it would be interesting how much the approach is able to derive from movement and how much from appearance in the visual modality.

- It is interesting how the approach scales to n-classes in Figure 3 but how does it scale to n-tasks? In the appendix there is an investigation into old classes but an investigation of the performance measures for, say, one chosen class after 4,5,6,7... tasks would be interesting to judge the trend.

**Limitations:**

The limitations are properly listed and the authors are upfront about them.

---

> ### Author Rebuttal · Authors · 2024-08-05
>
> Thank you for your valuable comments and encouraging remarks! We address your questions below. If there are any additional questions, we are happy to address them and revise our paper.
>
> > ### **Q1: Direct comparison between sound separation.**
>
> Thank you for your comment! Indeed, our paper focuses on developing continual learning methods to address catastrophic forgetting in audio-visual sound separation, rather than introducing new separation models. Our research demonstrates that existing audio-visual sound separation models suffer from catastrophic forgetting. We explored two separators, including the state-of-the-art iQuery model, and found that without advanced continual learning mechanisms, performance degrades significantly (e.g., SDR drops to 3.46, while our method achieves 7.33).
>
> Therefore, we believe a fair comparison involves applying various continual learning methods to the same separator model, as directly comparing different models without continual learning would result in significantly lower separation performance. This approach allows us to isolate the impact of continual learning techniques on the task. In future work, we will also explore developing a more general and robust audio-visual sound separation model architecture that inherently mitigates catastrophic forgetting.
>
>
> > ### **Q2: Figure 2 is too small.**
>
> Thanks for the suggestion! We have found more space for Figure 2 and enlarged it, by moving the dataset details into the Appendix.
>
>
> > ### **Q3: More detailed contrast to existing work in related work.**
>
> Thank you for your suggestion! We have added more details of existing works in the related work, such as "...Park et al. [47] extend the knowledge distillation-based [2, 15] continual image classification method to the domain of video by proposing the time-channel distillation constraint. Douillard et al. [14] proposed to tackle the continual semantic segmentation task with multi-view feature distillation and pseudo-labeling...". Due to space constraints, we cannot include the entire section here.
>
>
> > ### **Q4: All tasks have distinct classes. Is it a special case? Do you have an intuition or looked into variations where actually a smaller part of instances from a specific class is part of all tasks?**
>
> No, this is not a special case. In continual learning, models are typically trained on a sequence of *new* tasks with distinct classes. This setup is standard for evaluating a method's ability to tackle catastrophic forgetting.
>
> In scenarios where a smaller subset of instances from a specific class appears across all tasks, the forgetting issue for that specific class would be minimal or nonexistent. However, our proposed methods could still be applied to mitigate catastrophic forgetting in other classes and leverage knowledge from previous tasks to enhance fine-tuning on the shared classes in new tasks.
>
>
> > ### **Q5: Are there any compromises due to the hardware limitation?**
>
> Great question! We did not compromise the original data quality, such as image resolution or compression. Since the video encoder, object detector, and image encoder are frozen, we can pre-extract these features offline and use them to train the subsequent parts of the model, which allows us to remove the computational load of these three frozen components from the GPU, enabling the training process on a single RTX A5000 GPU with 24GB of memory.
>
> > ### **Q6: What is the memory size? How does the memory and computation scale to new tasks as the memory keeps growing beyond 4 tasks?**
>
> Great question! The memory size is set to 1 sample per old class. By keeping only 1 sample for each old class in memory, it becomes easy to scale to new tasks.
>
> > ### **Q7: Why choose 11kHz as the audio sampling rate?**
>
> We choose 11kHz as the audio sampling rate as it is a common setting for the MUSIC-21 dataset in existing audio-visual sound separation papers [1,2,3].
>
> > ### **Q8: How does this scale to hard examples, where both instruments are the same? How much the approach is able to derive from movement and how much from appearance in the visual modality?**
>
> In our model architecture iQuery [1], it consists of a spatial-temporal video encoder to extract the motion information from the given video. This allows the model to differentiate between instruments based on their motion patterns, even when they are from the same category and have similar audio characteristics.
>
> In hard cases where instruments have a similar appearance, the separation capability primarily relies on motion cues from the visual modality. However, when instruments have distinct appearances, the visual modality's appearance features can provide sufficient visual cues to effectively guide sound separation.
> > ### **Q9: How does Figure 3 scale to n-tasks? One chosen class after following tasks.**
>
> Nice suggestion! We randomly select one class ("accordion") from the first task and report its performance after training for each task. The results are shown in the following table (please see the following Official Comment with the title of "Results table of question Q9: How does Figure 3 scale to n-tasks? One chosen class after following tasks"), in which we can see that our method has an overall better performance compared to baselines. We can also observe that with incremental step increases, the performance of each method on this class tends to improve. This is because the memory data can be paired with data from previously unseen new classes to acquire new knowledge for old classes in subsequent tasks, as discussed in our paper. However, the performance drop compared to the upper bound still exists, demonstrating that catastrophic forgetting still occurs.
>
> [1] iQuery: Instruments as Queries for Audio-Visual Sound Separation. In *CVPR* 2023.
>
> [2] The Sound of Pixels. In *ECCV* 2018.
>
> [3] Co-Separating Sounds of Visual Objects. In *ICCV* 2019.

---

> ### Author Response · Authors · 2024-08-05
> **Results table of question Q9: How does Figure 3 scale to n-tasks? One chosen class after following tasks.**
>
> | Method              | step 1 |        |        | step 2 |        |        | step 3 |        |        | step 4 |        |        |
> |---------------------|--------|--------|--------|--------|--------|--------|--------|--------|--------|--------|--------|--------|
> |                     | SDR    | SIR    | SAR    | SDR    | SIR    | SAR    | SDR    | SIR    | SAR    | SDR    | SIR    | SAR    |
> | LwF                 | 3.65   | 6.16   | 12.54  | 4.97   | 7.53   | 10.50  | 4.09   | 9.68   | 6.90   | 4.50   | 10.19  | 7.42   |
> | PLOP                | 3.65   | 6.16   | 12.54  | 4.22   | 6.85   | 9.92   | 4.93   | 10.16  | 7.70   | 5.79   | 11.31  | 8.01   |
> | EWF                 | 3.65   | 6.16   | 12.54  | 2.62   | 4.42   | 9.71   | 4.02   | 9.16   | 7.84   | 5.01   | 11.03  | 7.37   |
> | ContAV-Sep (Ours)   | 4.46   | 6.80   | 11.86  | 5.14   | 6.60   | 13.09  | 6.62   | 10.49  | 9.89   | 6.02   | 10.15  | 10.05  |
> | Upper bound         | 3.65   | 6.16   | 12.54  | 6.84   | 9.44   | 11.73  | 7.68   | 11.37  | 11.83  | 10.43  | 13.99  | 13.89  |

---

> > ### Comment · Reviewer_49mS · 2024-08-10
> >
> > I have read the author's responses to my and the other reviewer's questions. Due to its focus on continual learning and added investigations of other datasets and answers to the questions I am confident in my rating.

---

> > > ### Author Response · Authors · 2024-08-10
> > > **Official Comment by Authors**
> > >
> > > Thank you so much for your positive support!

---

### Official Review · Reviewer_NCtq · 2024-07-23

**Soundness:** 2
**Presentation:** 2
**Contribution:** 2
**Rating:** 4
**Confidence:** 4

**Summary:**

This work proposes a continual audio-visual sound separation framework to mitigate the catastrophic forgetting problem

**Strengths:**

As I learned, this is the first work that focuses on the catastrophic forgetting problem in audio-visual separation task.

**Weaknesses:**

1. The font size in Figure 2 is too small, reducing the readability of this paper.
2. The technique novelty is limited, the model architecture is totally the same as iQuery, and the most brightness point of this work is just proposes a cross-modal similarity distillation constraint, however, it just the combination of the contrastive loss implemented on the modalities and features extracted from different training step.
3. Experiments are not enough, all experiments are conducted on Music21, which just contains limited data among 21 classes. Experiments conducted on Music [1], VGGSound [4], and AVE datasets [2-3] can provide a more comprehensive evaluation.
[1] The sound of pixels.
[2] Audioset: An ontology and human-labeled dataset for audio events.
[3] Audio-visual event localization in unconstrained videos
[4] Vggsound: A large-scale audio-visual dataset

**Questions:**

Please seem the weakness part.

**Limitations:**

1. Limited novelty.
2. Insufficient experiments

---

> ### Author Rebuttal · Authors · 2024-08-05
>
> Thank you for your valuable comments and suggestions! We address the raised concerns below. If there are any additional questions, we are willing to address them and revise our paper.
>
> > ### **Q1: The font size in Figure 2 is too small.**
>
> Thank you for your suggestion! We have enlarged the font size in Figure 2.
>
>
> > ### **Q2: Same architecture as iQuery.**
>
> In our work, we address the proposed continual audio-visual sound separation problem, focusing on the challenge of **training audio-visual sound separation models continuously while mitigating the catastrophic forgetting typically associated with continual learning**. To benchmark and fairly compare with existing continual learning approaches, we do not introduce new separation architectures. Instead, we concentrate on developing new learning approaches based on current state-of-the-art separators, i.e., iQuery [1], to tackle the catastrophic forgetting problem in separation, aligning with similar research goals in the broader continual learning literature.
>
> Furthermore, besides iQuery [1], in Table 1 of Section 4.2, we present experimental results based on another model architecture, Co-Separation [2]. Our method outperforms baseline methods in this context as well, demonstrating the generalizability of our proposed approach to different separation model architectures.
>
>
> > ### **Q3: Cross-modal Similarity Distillation Constraint is just the combination of the contrastive loss implemented on the modalities and features extracted from different training step.**
>
> We would like to note that the proposed Cross-modal Similarity Distillation Constraint (CrossSDC) is a non-trivial contribution. CrossSDC is the first method to integrate previously learned features into the contrastive learning process of the current task. This enables maintaining cross-modal semantic similarity incrementally while distilling prior knowledge from older tasks directly into the current model. This simple yet effective unified training target/constraint effectively addresses catastrophic forgetting in our Continual Audio-Visual Sound Separation task. Extensive experiments with two different separation models validate its effectiveness.
>
> > ### **Q4: Experiments conducted on other datasets can provide a more comprehensive evaluation.**
>
> Thank you for suggesting additional datasets such as Music, VGGSound, and AVE! Following common practice in recent audio-visual sound separation works, we use the Music-21 dataset as our main benchmark. Note that Music [3] is a smaller subset of Music-21. To further validate our method, we conducted experiments on the AVE dataset and have included the results in the Appendix. The results for Fine-tuning/LwF/PLOP/AV-CIL/ContAV-Sep (Ours) are 2.07/2.19/2.45/2.53/**2.72** for SDR and 5.64/6.43/6.11/6.64/**7.32** for SIR, respectively, which further demonstrate the effectiveness of our approach. For more details, please refer to Section A.3 in the Appendix. We will move these results to the main paper. Additionally, experiments on larger datasets such as VGGSound and AudioSet would be very interesting for future work, and we will discuss these in the main paper.
>
>
> [1] iQuery: Instruments as Queries for Audio-Visual Sound Separation. In *CVPR* 2023.
>
> [2] Co-Separating Sounds of Visual Objects. In *ICCV* 2019.
>
> [3] The Sound of Pixels. In *ECCV* 2018.

---

> > ### Author Response · Authors · 2024-08-12
> > **Official Comment by Authors**
> >
> > Dear Reviewer NCtq
> >
> > We sincerely appreciate your valuable feedback, which has greatly improved our paper! We kindly ask if you could confirm whether our response has adequately addressed your concerns. If so, we would be grateful if you might consider raising your rating. Please do not hesitate to let us know if there are any remaining issues.
> >
> > Thank you once again for your insightful feedback!
> >
> > Best Regards,
> >
> > The Authors

---

> > > ### Comment · Reviewer_NCtq · 2024-08-12
> > > **Reply**
> > >
> > > Thank you to the authors for their response. However, I still have concerns regarding the technical contribution and the adequacy of the experiments conducted.
> > >
> > > **Limmited Technical Contribution**: I remain unconvinced about the significance of the technical contribution in this paper. As the authors mentioned, they employ iQuery as their primary framework to address the issue of catastrophic forgetting in audio separation. If we consider the overview in the manuscript as a simplified introduction to iQuery's workflow, it suggests that the framework is not novel. The authors argue that the Cross-modal Similarity Distillation Constraint (CrossSDC) is their main contribution, highlighting it as the first method to integrate previously learned features into the contrastive learning process for the current task. However, I find this claim unconvincing because similar techniques have been applied to other tasks in previous work [1][2]. This makes the contribution seem more like an extension for achieving the audio speparation rather than a novel innovation.
> > >
> > > **Insufficient Experimental demonstration**: There are also concerns regarding the sufficiency of the experimental validation. While [3][4] conduct audio separation experiments on VGGsound, Music21 only includes 21 categories, and the AVE dataset used by the authors comprises around 20 categories. This limited scope raises doubts about whether the experiments are comprehensive enough to convincingly demonstrate the method's effectiveness, as the author emphasized they focus on the catastrophic forgetting problem. I would be more inclined to improve my evaluation if the method's effectiveness were demonstrated across a broader range of data categories."
> > >
> > > [1] Distilling Audio-Visual Knowledge by Compositional Contrastive Learning
> > > [2 ]Learning from Students: Online Contrastive Distillation Network
> > > [3] Language-Guided Audio-Visual Source Separation via Trimodal Consistency
> > > [4] A Unified Audio-Visual Learning Framework for Localization, Separation, and Recognition

---

> > > > ### Author Response · Authors · 2024-08-12
> > > > **Response to Reviewer NCtq**
> > > >
> > > > We thank your further response to our rebuttal and your willing to revise your evaluation based on the results on the dataset with larger number of categories.
> > > >
> > > > > ### **Regarding the contribution**
> > > >
> > > > We would like to further emphasize our contributions. Beyond our proposed approach, the Continual Audio-Visual Sound Separation task itself is a significant contribution of our paper.
> > > >
> > > > In the approach, our technical contribution is not to introduce a separator framework, such as iQuery, for sound separation. Instead, similar to the research goals in the broader continual learning literature, where the main concentration is to explore **"how to mitigate the catastrophic forgetting probelm when training the model continually across tasks?"**, the primary research direction in this paper is to address the catastrophic forgetting problem when continually training the audio-visual sound separation model. Further, our technical contribution can be seamlessly adapted to different model architectures for Continual Audio-Visual Sound Separation. The experimental results in Table 1 demonstrate that our approach consistently outperforms baseline methods, even when applied to other architectures like Co-Separation [1], highlighting the generalizability of our approach across various separation models.
> > > >
> > > > While the knowledge distillation methods in [2, 3] have some similar characteristics with our CrossSDC, there are key differences that set our approach apart. Unlike the method in [2], which employs a separate distillation loss to align the prediction distribution/logits across different modalities and classes, and unlike the approach in [3], which uses contrastive loss to ensure class similarities between teacher and student models without preserving any cross-modal semantic similarities, our CrossSDC construct an unified cross-task cross-modal contrastive loss by integrating previously learned audio and visual features into the computing process of cross-modal contrastive learning, allowing the cross-modal similarity calculation process across different tasks, enabling not only the distilation of previous learned cross-modal knowledges/similarities but also the enhancement of the learning of current task's audio-visual similarities. We have cited and discussed these papers in our revised version.
> > > >
> > > >
> > > > > ### **Experiments on a broader range of sound categories**
> > > >
> > > > To demonstrate that the approach can handle a broad range of categories, we have conducted experiments on the VGGSound dataset, in which we use randomly selected 100 categories for our continual audio-visual sound separation. We train both our method and the baseline methods using iQuery as the separator architecture. The results for Fine-tuning/LwF/PLOP/AV-CIL/ContAV-Sep (Ours) are 3.69/4.71/4.56/4.66/**4.90** for SDR and 7.23/8.89/8.32/8.61/**9.25** for SIR, respectively, which further demonstrate the effectiveness of our approach on a broader range of data categories.
> > > >
> > > >
> > > > [1] Co-Separating Sounds of Visual Objects. In *ICCV* 2019.
> > > >
> > > > [2] Distilling Audio-Visual Knowledge by Compositional Contrastive Learning. In *CVPR* 2021.
> > > >
> > > > [3] Learning from Students: Online Contrastive Distillation Network for General Continual Learning. In *IJCAI* 2021.

---

> ### Comment · Reviewer_NCtq · 2024-08-12
> **Reply**
>
> Many thanks to the reviewers for their responses, I was surprised by how quickly the authors downloaded and processed the 100-class VGGSound dataset from YouTube. The 100 classes may contain roughly 90K videos. I was also surprised by the training speed of the authors. However, I am still not confident enough to recommend this paper for acceptance, **mainly because of its low technical contribution.** I will maintain my original score.

---

> > ### Author Response · Authors · 2024-08-12
> > **Response to Reviewer NCtq**
> >
> > Thank you for your prompt response! We’d like to clarify that we began experiments with VGGSound at the end of July after receiving the reviewer’s feedback. However, due to the time required for data preparation and model training, we couldn’t provide results earlier. We initially believed the results on MUSIC-21 and AVE sufficiently validated our findings.
> >
> > **We also want to highlight that the two works mentioned by the reviewer utilize only small subsets of VGGSound and AudioSet, as described in their papers or appendices.** For example, [3] uses only 15 musical instrument categories from AudioSet, and [4] focuses on 49 music categories from VGGSound-Music (even though the full set was adopted in this work for sound source localization task and their separation task only used this small subset).  **In contrast, our VGGSound subset consists of 100 significantly more broad and diverse categories, extending beyond just musical instruments.** We believe these results clearly strengthen our contributions.
> >
> > We will release all of our source code and pre-trained models. While we respectfully disagree with your assessment of the technical contribution, we leave the final judgment to you, the other reviewers, and the AC. We appreciate your perspective and value your comments and engagement in the discussions. Your suggestion on a broader range of data classes has indeed made our paper stronger.
> >
> > [3] Language-Guided Audio-Visual Source Separation via Trimodal Consistency. In *CVPR* 2023.
> >
> > [4] A Unified Audio-Visual Learning Framework for Localization, Separation, and Recognition. In *ICML* 2023.

---

> ### Author Response · Authors · 2024-08-12
> **More details of the experiments on VGGSound**
>
> We’d like to provide more details about our VGGSound experiments to address the reviewer’s doubts.
> We randomly selected the following 100 categories:
>
> ['playing theremin', 'donkey, ass braying', 'playing electronic organ', 'zebra braying', 'people eating noodle', 'airplane flyby', 'playing double bass', 'cat growling', 'footsteps on snow', 'playing tennis', 'black capped chickadee calling', 'bouncing on trampoline', 'playing steelpan', 'waterfall burbling', 'subway, metr', 'people clapping', 'chipmunk chirping', 'chopping food', 'people shuffling', 'elk bugling', 'alarm clock ringing', 'people booing', 'canary calling', 'chopping wood', 'people humming', 'lathe spinning', 'playing tuning fork', 'playing violin, fiddle', 'singing choir', 'playing timbales', 'children shouting', 'chicken crowing', 'car passing by', 'driving motorcycle', 'bull bellowing', 'lawn mowing', 'playing bugle', 'mouse squeaking', 'child singing', 'playing tympani', 'hair dryer drying', 'basketball bounce', 'driving snowmobile', 'train whistling', 'thunder', 'dog bow-wow', 'ocean burbling', 'cuckoo bird calling', 'sheep bleating', 'splashing water', 'air conditioning noise', 'cattle mooing', 'eagle screaming', 'air horn', 'playing bass guitar', 'sloshing water', 'tap dancing', 'running electric fan', 'playing ukulele', 'playing guiro', 'playing shofar', 'people sniggering', 'people whispering', 'people finger snapping', 'car engine idling', 'bathroom ventilation fan running', 'police car (siren)', 'roller coaster running', 'playing french horn', 'swimming', 'lighting firecrackers', 'playing electric guitar', 'playing castanets', 'people babbling', 'arc welding', 'wood thrush calling', 'wind rustling leaves', 'playing darts', 'planing timber', 'crow cawing', 'shot football', 'writing on blackboard with chalk', 'people slapping', 'using sewing machines', 'raining', 'dog howling', 'playing cello', 'playing trumpet', 'fox barking', 'bowling impact', 'people crowd', 'pumping water', 'ice cracking', 'baby crying', 'playing bass drum', 'playing bongo', 'tornado roaring', 'playing steel guitar, slide guitar', 'playing squash', 'typing on typewriter'].
>
> As you can see, our dataset is diverse, containing not only musical instruments but also human sounds, sports, traffic, animals, and various other categories. To the best of our knowledge, recent state-of-the-art methods, including the two suggested by the reviewer, have rarely been tested on such a diverse range of categories in the context of audio-visual sound separation.
>
> The total number of videos in this subset is 61,195. For each category, we randomly selected 20 videos for validation, 20 videos for testing, and the remainder for training. This results in 57,195 videos for training, 2000 mixtures for validation, and 2000 mixtures for testing. We divided the 100 categories into 4 incremental tasks, each containing 25 categories. Both our method and the baseline methods were trained for 100 epochs per task.
>
> We hope these details further demonstrate the breadth and rigor of our experiments on VGGSound, strengthening our contributions.

---

### Author Response · Authors · 2024-08-09
**Follow-up on the Rebuttal**

Dear Reviewers,


Thank you for all of the valuable comments and suggestions provided in the reviews. We have addressed your questions in our rebuttal and would appreciate it if you could take a look and share any further questions or suggestions you may have. We’re happy to address them!


Best regards,

Paper 3807 Authors

---

### Comment · Area_Chair_B8hC · 2024-08-12
**Author-reviewer discussion ending soon**

Dear authors and reviewers,

Thank you for the robust discussions so far.

Reviewers 5v9b and wmVP: it would be very valuable to have your input to the discussion, and the authors have posted detailed responses to your reviews. Could you please respond to acknowledge that you have read and considered the reviews and author rebuttal, and post any remaining questions that would be useful to discuss with the authors? We only have about a day left in the discussion period, so please prioritize this.

Thanks in advance.

All the best,
Your AC

---

### Author Response · Authors · 2024-08-12
**Thank you all for your time and consideration**

Dear Area Chair and Reviewers,

Thank you all for your time and effort in reviewing our submission!

We are sincerely grateful for the constructive feedback provided by all reviewers, which has significantly improved our work. We also appreciate the insightful discussions and the professional and respectful manner in which they were conducted throughout the rebuttal and discussion process. We have taken care to ensure our responses respect the opinions of all reviewers.

While we understand that some concerns may remain, we hope our explanations have addressed them satisfactorily. We remain committed to providing any further clarification or addressing additional questions that may arise.

Thank you again for your time and consideration!

Best regards,

The Authors of Paper 3807

---

### Decision · Program_Chairs · 2024-09-25

**Decision:**

Accept (poster)

**Comment:**

This paper proposes the continual audio separation task. The goal of this task is to adapt an audio-visual separation model to new source classes while maintaining separation performance on previously-learned tasks, in a continual fashion. The proposed approach uses the iQuery model as a foundation, and adds a "cross-modal similarity distillation constraint (CrossSDC)", which enforces both instance-aware and class-aware cross-modal similarities through a combination of contrastive loss and knowledge distillation, which prevent catastrophic forgetting. The approach is evaluated on MUSIC21, and after rebuttal, also VGGSound.

The majority of reviewers recommend at least borderline accept, especially after the author rebuttal and discussion. Particular strengths mentioned include:

1. Clearly written paper that is easy to understand, with good motivation for the problem addressed (continual audio-visual separation) and for the proposed solution (CrossSDC to prevent catastrophic forgetting)

2. Demos in the included supplementary are impressive and convincing.

3. Thorough evaluation and analysis of results, including ablations, especially of memory size.

Weaknesses mentioned include the following, along with corresponding

1. Concern about novelty, as the method is based on the prior iQuery method, and also that similar contrastive methods to the proposed CrossSDC have been used in potentially similar contexts before. The most critical reviewer (NCtq) considers the combination of these two prior works and the application of CrossSDC to the related task of continual audio-visual separation to fall short as a significant contribution. I agree this isn't a completely novel contribution, but to me (and other reviewers) it seems that the application to a new task is interesting and novel enough.

2. Inital concern about lack of generalization, as the initial results were only on MUSIC21. The authors presented results on more general VGGSound data in the rebuttal, which convinced reviewers that the method is more generally applicable.

3. Some lack of discussion of prior works, e.g. audio-only continual separation works.

Given that the majority of reviewers find value in this work, especially after the author rebuttal and dsicussion, I think it is over the bar of acceptance, and an interesting contribution to the areas of audio-visual source separation and continual learning.